# Simulating the effects of mobility restrictions in the spread of SARS-CoV-2 in metropolitan areas in Portugal

**Sandra Oliveira**[1,2]\*, **Ana Isabel Ribeiro**[3,4,5], **Paulo Nogueira**[6,7,8,9], **Jorge Rocha**[1,2]

**1** Centre for Geographical Studies, Institute of Geography and Spatial Planning, Universidade de Lisboa, Lisbon, Portugal, **2** Associated Laboratory Terra, Lisbon, Portugal, **3** EPIUnit, Instituto de Saúde Pública da Universidade do Porto, Porto, Portugal, **4** Laboratório para a Investigação Integrativa e Translacional em Saúde Populacional (ITR), Faculdade de Medicina da Universidade do Porto, Porto, Portugal, **5** Departamento de Ciências da Saúde Pública e Forenses e Educação Médica, Faculdade de Medicina, Universidade do Porto, Porto, Portugal, **6** IMPSP—Instituto de Medicina Preventiva e Saúde Pública, Faculdade de Medicina, Universidade de Lisboa, Lisbon, Portugal, **7** Área Disciplinar Autónoma de Bioestatística (Laboratório de Biomatemática), Faculdade de Medicina, Universidade de Lisboa, Lisbon, Portugal, **8** Instituto de Saúde Ambiental, Faculdade de Medicina, Universidade de Lisboa, Lisbon, Portugal, **9** EPI Task-Force FMUL, Faculdade de Medicina, Universidade de Lisboa, Lisbon, Portugal

\* sandra.oliveira1@campus.ul.pt

**Data Availability Statement:** The data used in this study are owned by third-party organizations and are available elsewhere, from the repositories or websites of the corresponding owner entities, as

## Abstract

Commuting flows and long-distance travel are important spreading factors of viruses and particularly airborne ones. Therefore, it is relevant to examine the association among diverse mobility scenarios and the spatial dissemination of SARS-CoV-2 cases. We intended to analyze the patterns of virus spreading linked to different mobility scenarios, in order to better comprehend the effect of the lockdown measures, and how such measures can be better informed. We simulated the effects of mobility restrictions in the spread of SARS-CoV-2 amongst the municipalities of two metropolitan areas, Lisbon (LMA) and Porto (PMA). Based on an adapted SEIR (Suscetible-Exposed-Infected-Removed) model, we estimated the number of new daily infections during one year, according to different mobility scenarios: restricted to essential activities, industrial activities, public transport use, and a scenario with unrestricted mobility including all transport modes. The trends of new daily infections were further explored using time-series clustering analysis, using dynamic time warping. Mobility restrictions resulted in lower numbers of new daily infections when compared to the unrestricted mobility scenario, in both metropolitan areas. Between March and September 2020, the official number of new infections followed overall a similar timeline to the one simulated considering only essential activities. At the municipal level, trends differ amongst the two metropolitan areas. The analysis of the effects of mobility in virus spread within different municipalities and regions could help tailoring future strategies and increase the public acceptance of eventual restrictions.

listed below (and mentioned in the manuscript): -Data on mobility patterns between municipalities of metropolitan areas are owned by National Statistics Portugal (INE-Instituto Nacional de Estatística) and although they are public data, there are restrictions and the institute demands that users send a special request, to ensure anonymity and proper use of the data. The request can be done at this link: https://www.ine.pt/xportal/xmain?xpid=INE&xpgid=ine_ped_informacao - Data on the Effective Reproduction Number (Rt) is freely available from the National Institute Ricardo Jorge, Ministry of Health (INSA, in Portuguese), and can be accessed via this link: http://www.insa.min-saude.pt/category/areas-de-atuacao/epidemiologia/covid-19-curva-epidemica-e-parametros-de-transmissibilidade -Data on SARS-CoV-2 cases in Portugal are available from the Directorate-General of Health (DGS, in Portuguese), from a specific website: https://covid19.min-saude.pt/relatorio-de-situacao/, and are presented only in pdf format. To enable the usability of these data for different applications, the organization Data Science for Social Good Portugal (https://www.dssg.pt/) has created a free repository to store and distribute these data, available at: https://github.com/dssg-pt/covid19pt-data.

**Funding:** This work was financed by national funds through FCT—Portuguese Foundation for Science and Technology, I.P., under the framework of the project "TRIAD - health Risk and social vulnerability to Arboviral Diseases in mainland Portugal" [PTDC/GES -OUT/30210/2017] and by the Research Unit UIDB/00295/2020 and UIDP/00295/2020. This study was also funded by FEDER through the Operational Programme Competitiveness and Internationalization and national funding from the Foundation for Science and Technology – FCT (Portuguese Ministry of Science, Technology and Higher Education) under the EPIUnit (UIDB/04750/2020). AIR was supported by National Funds through FCT, under the programme of 'Stimulus of Scientific Employment – Individual Support' within the contract CEECIND/02386/2018. The funders had no role in study design, data collection and analysis, decision to publish, or preparation of the manuscript.

**Competing interests:** The authors have declared that no competing interests exist.

# 1. Introduction

The outbreak of the SARS-CoV-2 virus started in December 2019 in Wuhan, China, and has disseminated rapidly throughout the world. Many countries have implemented strict lockdown procedures to stop the spread of infections and imposed limitations on mobility and social contacts, such as traffic control and the temporary suspension of all non-essential travel [1, 2]. In the USA, a mobility reduction between 40 and 60% due to SARS-CoV-2 restrictions was observed [3]. In Europe, early results show that mobility was sharply reduced in response to the first lockdowns and increased again when restrictions were lifted [4]. In the Netherlands, a decrease in the number of trips (55% reduction) and distance traveled (68% reduction) was observed [5]; and in France, there were 65% fewer trips [6], similar to the reduction observed in Italy [7].

Prior studies have shown that commuting and long-distance travel were important spreading factors of viruses, especially airborne ones [8–11]. A previous study in Sweden indicated that banning travels over 50 km distance could considerably decrease the spatiotemporal spread of disease outbreaks, such as SARS (Severe Acute Respiratory Syndrome), even when considering that not all the population would comply [12]. In such a context, mobility restrictions are considered efficient approaches to decrease the spread of a virus [13], and their effects on the dissemination of SARS-CoV-2 are being investigated. In Italy, Cartenì et al. [14] found a positive correlation between transport accessibility and COVID-19 cases, using a multiple linear regression model. In the USA, using a generalised linear model, Badr et al. [15] observed that a reduction in mobility flows at the county level was strongly correlated with a decrease in COVID-19 cases, even though this effect was delayed for up to 3 weeks. Other studies have also tested the relation of COVID-19 spread with the implementation of control measures using epidemiological models, such as SIR or S(E)IR (Susceptible, (Exposed), Infected, Recovered/Removed), widely used in epidemiological studies [16, 17]. In Germany, Dehning et al. [18] estimated up to 40% decrease in virus spreading rate during March 2020, linked to governmental interventions that limited social contact. Likewise, Gatto et al. [19] estimated a 45% reduction in virus transmission in Italy due to restrictions in mobility and human-to-human interactions, between February and March 2020.

In Portugal, the first control measures were applied on 12[th] March 2020, 10 days after the first COVID-19 cases were recorded; on 18[th] March, the state of emergency was declared, and a strict lockdown was implemented ([20]; DR 55/2020). Internal traveling was limited to essential services, remote work was declared compulsory and schools were closed until early May 2020. Due to a low number of fatalities in this early stage of the pandemic, Portugal was portrayed as a successful case of COVID-19 control [21]. Throughout the year 2020, other lockdown periods were implemented, linked to the dynamics of the virus spread in the country. In some cases, particularly restrictive measures to inter-municipal mobility were applied to reduce the flow of people expected in specific holidays. Early findings point out to an evident decrease in virus dissemination rate due to the reduction in mobility levels, as measured by mobile positioning data [22], although the paths of virus spreading reveal a heterogeneous pattern among municipalities [20].

This study intended to investigate the relationship between different mobility scenarios and the spatial dissemination of SARS-CoV-2 cases. Using virus spread simulations based on a SEIR model, we estimate the potential effects of the flow of people, according to specific commuting conditions and restrictions, in the number of COVID-19 cases and their spatial distribution. The analysis was applied to the metropolitan areas of Lisbon and Porto in Portugal. Composed by a set of adjacent and interconnected municipalities, metropolitan regions constitute the most densely populated areas in the country, with a high concentration of economic

activities, intense commuting flows amongst the different municipalities and a diversified transport network. We aimed to investigate the patterns of virus spread potentially linked to mobility conditions in these areas, to further understand the influence of the lockdown measures implemented in the virus dissemination, and how such measures could be better adjusted in the future.

## 2. Materials & methods

### 2.1. Study areas

**2.1.1. Lisbon Metropolitan Area (LMA).** LMA is located in the west-central part of Portugal, centered on the mouth of the river Tagus. It is composed of 18 municipalities, 9 of them located in the north side of Tagus River, and 9 in the south, and includes Lisbon's capital city (Fig 1). It extends over 3015 km$^2$, with the biggest municipality having an area of 465 km$^2$ (Palmela) and the smallest 24 km$^2$ (Amadora). In 2019, the resident population in LMA reached 2 840 005 inhabitants [23], which corresponds to 27.6% of the country's total population. The most populated municipality is Lisbon, with 506 654 inhabitants, whereas Alcochete has the lowest number of residents (19 395). Nearly 81% of the population commutes, predominantly by car (59%), whereas collective public transport (composed of train, metro, bus, and boat) is used by 16% of the commuters [24].

Intermunicipal flows reveal that Lisbon is the largest receiver, mainly from Sintra, Amadora, Odivelas, Loures, Oeiras, and Vila Franca de Xira in the north side of Tagus River, and from Almada and Seixal in the south (Fig 2). These are also the municipalities with the largest share of public transport flow to Lisbon, which receives nearly half of the people that commute by public transport within the metropolitan area. When considering only the essential activities' flows, these patterns change, and people commuting are dispersed through more municipalities, including Palmela and Setúbal in the south, which in the other mobility scenarios represent less than 3% of the flows.

**2.1.2. Porto Metropolitan Area (PMA).** PMA is located in the northwest part of Portugal, centered on the mouth of the river Douro. It is composed of 17 municipalities, 10 of them located in the north side of the Douro river, and 7 in the south, including the second most populated city of Portugal (Porto) (Fig 1). PMA extends over 2041 km$^2$, with the most extensive municipality having an area of 329 km$^2$ (Arouca) and the smallest 8 km$^2$ (São João da Madeira, SJMadeira). In 2019, the resident population was 1 721 038 [23], which corresponds to 16.7% of the country's total population. The most highly populated municipality is Vila Nova de Gaia, with 299 879 people, whereas Arouca is the least populated (20 950). Nearly 79% of the population commutes, predominantly by car (68%). In contrast, collective public transport is used by only 11% of the mobile population of the metropolitan area [24].

Intermunicipal flows reveal that Porto is the largest receiver, mainly from Gondomar, Maia, Matosinhos, and Vila Nova de Gaia (Fig 3). These are also the municipalities with the largest share of public transport flow to Porto and Valongo. About a third of the population from Valongo and Gondomar commutes by public transport. Maia and Paredes receive commuters from the contiguous municipality of Valongo. The highest proportion of transport flows from Vila do Conde are directed to the neighbouring municipality of Póvoa de Varzim, except when considering public transport, with more people commuting to Porto using this mode. The flows regarding essential activities are more dispersed amongst the municipalities that compose PMA, with Santo Tirso and Oliveira de Azeméis representing a larger proportion of the flows when compared with the other mobility scenarios.

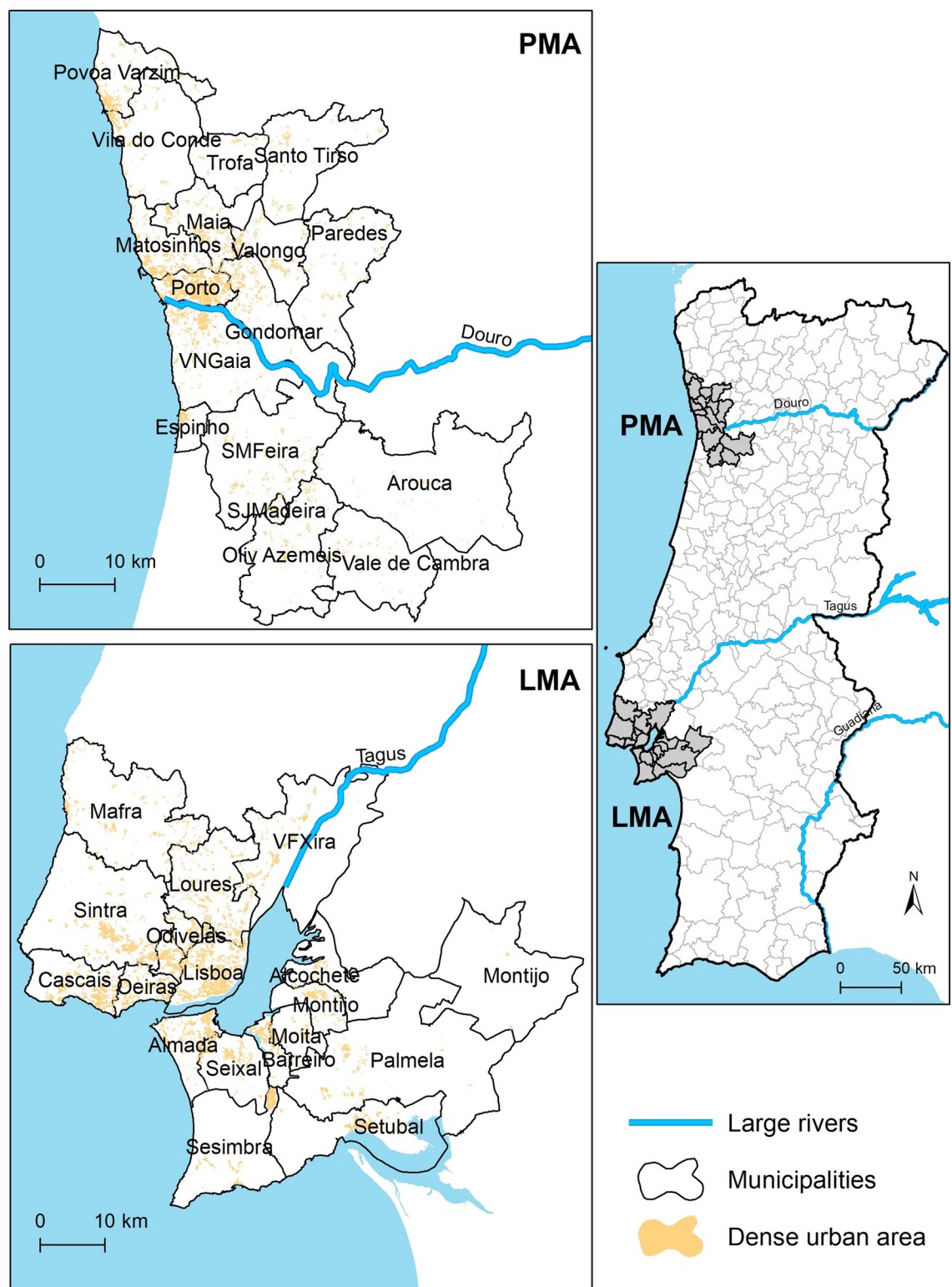

**Fig 1. Location of the two metropolitan areas of Portugal (LMA–Lisbon, PMA–Porto), with the distribution of the dense urban areas within their territory.**

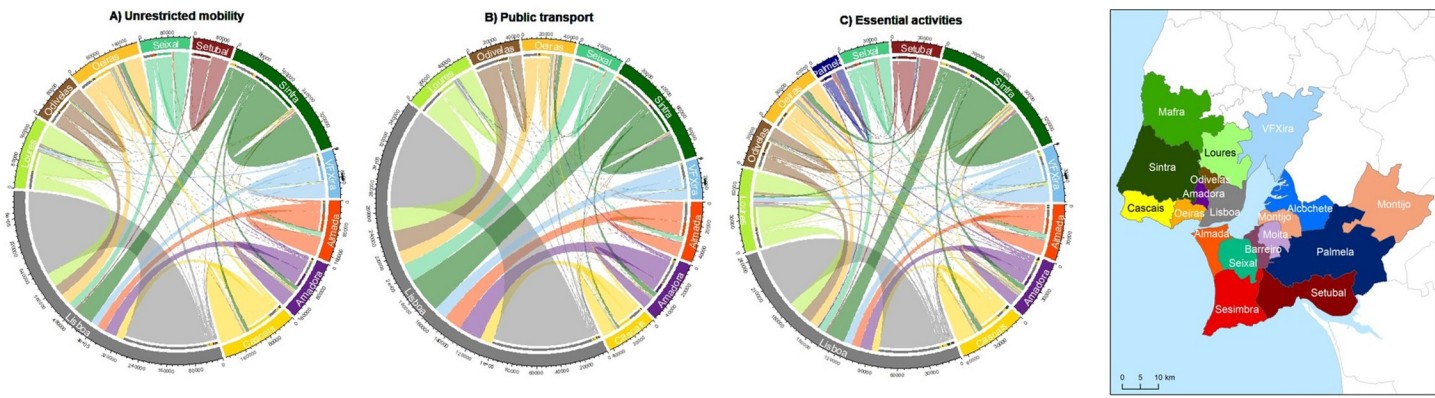

**Fig 2. Daily flows amongst LMA municipalities.** A) unrestricted mobility with all transport modes (data of 2011); B) mobility based on the use of collective public transport; C) mobility restricted to essential activities Origin is identified by the color of the corresponding municipality, shown in the map on the right. The degree of flows is represented by the width of the colored links. To facilitate visualization, flows representing less than 3% of the total were excluded. Graphs were made with circlize package in R software [25]. Longer names of municipalities were reduced for simplification: VFXira = Vila Franca de Xira. Source of data: Statistics Portugal (INE, 2011).

## 2.2. Data collection

**2.2.1. Mobility.** We collected statistical data on travel flows between the different municipalities for each metropolitan area, based on the latest available Census from Statistics Portugal (INE, Instituto Nacional de Estatística) [26]. These data represent the number of people who commute daily, for work or study, between an origin (where they live) and a destination (where they work or study), disaggregated by transport mode. To analyze distinct mobility patterns within the metropolitan areas, we combined the data to create different layers: 1) the total number of people travelling between municipalities using any transport mode (Table 1, 1); 2) the number of people travelling between municipalities using collective transport modes (train, metro, bus and boat, Table 1, 2). In addition, we collected statistical data regarding major sectors of activity based as well on the available data from the latest Census 2011 [26]. These data represent the number of people who work or study in different sectors of activity, disaggregated by origin (place of residence) and destination (where they work or study). The sectors of activity are divided into 21 categories, as presented in Table 1. Based on the

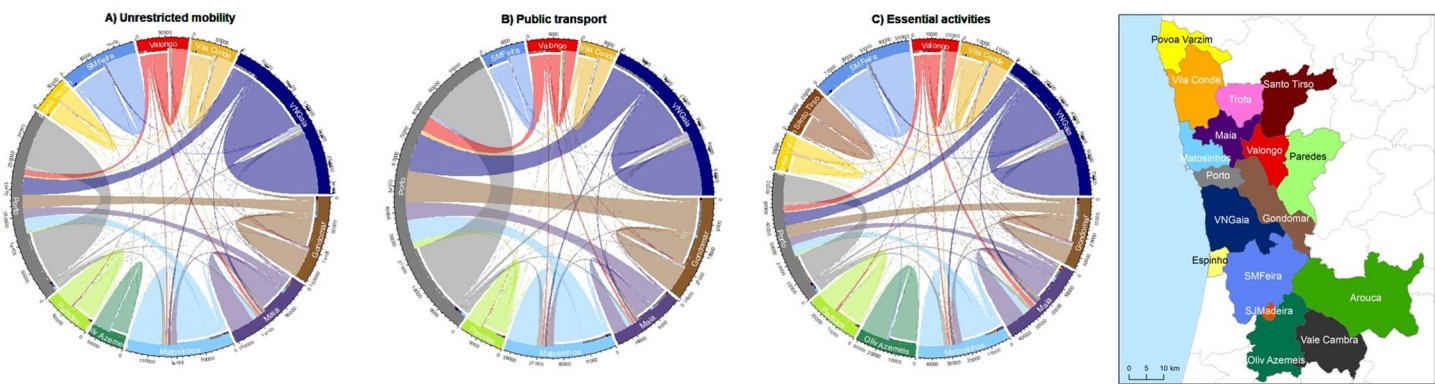

**Fig 3. Daily flows amongst PMA municipalities.** A) unrestricted mobility with all transport modes (data of 2011); B) mobility based on the use of collective public transport; C) mobility restricted to essential activities. Origin is identified by the color of the corresponding municipality, shown in the map on the right. The degree of flows is represented by the width of the colored links To facilitate visualization, flows representing less than 3% of the total were excluded. Graphs were made with circlize package in R software [25]. Longer names of municipalities were reduced for simplification: VNGaia = Vila Nova de Gaia; SMFeira = Santa Maria da Feira; SJMadeira = São João da Madeira; Oliv Azemeis = Oliveira de Azeméis.

**Table 1. Data collected regarding different mobility scenarios, for LMA and PMA.**

| Mobility scenarios | Data layer | Source | Characteristics |
|---|---|---|---|
| 1) Unrestricted Mobility (2011) | Number of people travelling between municipalities, all transport modes | INE, 2011 | Includes all transport modes (car, walk, bus, train, metro, boat, bike, other) |
| 2) Mobility based on the use of public/ collective transport | Number of people travelling between municipalities using public & collective transport | INE, 2011 | Includes public collective transport modes (bus, train, metro, boat) and company buses |
| 3) Mobility restricted to essential activities | Number of people travelling between municipalities who work in essential activities | INE, 2011 | Agriculture, fisheries, and forestry; extraction industry; transformation industry; electricity and gas supply; water supply, sanitation and residues; construction; trade and auto repair; transport and storage; human health |
| | Other activities, non-essential, temporarily suspended during lockdowns or done remotely | INE, 2011 | Accommodation and restaurants; information and communication; finance and insurance; real estate; scientific and technical; public administration and social security; education; arts and sports; other services; domestic; international organizations |
| 4) Mobility restricted to industrial activities | Number of people travelling between municipalities who work in industrial activities | INE, 2011 | Includes industrial activities, extraction and transformation |
| 5) Unrestricted Mobility (2017) | Number of people travelling between municipalities, all transport modes | INE, 2017 | Includes all transport modes (car, walk, bus, train, metro, boat, bike, other). |
| | | | Dataset is incomplete, with 14% missing data for LMA and 35% for PMA |

restrictions applied by the Portuguese government, we selected those sectors that were considered essential and were not suspended during lockdowns, specifically: Agriculture, fisheries, and forestry; extraction industry; transformation industry; electricity and gas supply; water supply, sanitation, and residues; construction; trade and auto repair; transport and storage; and human health. We then calculated the number of people working in these essential sectors and that commuted between the municipalities within each metropolitan area. Since all school activities were suspended during stricter lockdown periods, students were excluded. As such, this new layer (Table 1, 3) represents another mobility scenario, restricted to the people who commute to work regarding only essential activities. Another layer was also created, limited to the people who work solely in industrial activities (Table 1, 4), to test if any specific relationship exists with the patterns of virus dissemination. This option derived from prior studies on the SARS-CoV-2 dispersion, considering that industrial activities may require close interaction amongst workers [27] and a high contagion risk related to the presence of industrial activities has been found in Portugal, mainly in the northern region [28].

The data on transport modes and activity sectors are not connected. Therefore, it was not possible to organize the layers 3) and 4) (Table 1) according to transport mode. Since the focus was on patterns within metropolitan areas, travel to and from municipalities outside LMA or PMA was disregarded for all layers.

More recent data on mobility patterns within both metropolitan areas are available, obtained with a specific survey carried out in 2017 by Statistics Portugal [24]. However, these data are incomplete for most municipalities, with 14% of missing data for LMA and 35% for PMA. As such, although it may be useful to verify potential changes in commuting levels since 2011, we had to use these data with caution and restricted to the total number of people travelling between municipalities in all transport modes combined (Table 1, 5).

**2.2.2. Epidemiological data.** We have obtained the Effective Reproduction Number ($R_t$), available daily, for the period between 23rd March 2020 to 31st March 2021, from the National Institute Ricardo Jorge, Ministry of Health (INSA, in Portuguese, http://www.insa.min-saude.pt/category/areas-de-atuacao/epidemiologia/covid-19-curva-epidemica-e-parametros-de-transmissibilidade). $R_t$ is a derivation of $R_0$, the basic reproduction number that represents how each infected person produces many secondary infections. $R_t$ is the reproduction number

over time, or $R_0$ at time $_t$ [29, 30]. $R_t$ data are available at the regional level; therefore, we used the daily $R_t$ values of the North region for the simulations applied to PMA, and those available for the region of Lisbon and Tagus Valley for LMA simulations.

Data on the number of SARS-CoV-2 cases per municipality in Portugal became available since 23$^{rd}$ March 2020, provided by the Directorate-General of Health (DGS, in Portuguese). Until 6$^{th}$ July, the data were provided daily, with an update on the cumulative number of cases per municipality. From that day on and until the 26$^{th}$ October 2020, the data were provided only once a week. After that, the data started to be aggregated for 14 days to calculate an incidence rate (number of cases/100.000 residents). We collected the available data per municipality for each metropolitan area for the period under study (23$^{rd}$ March 2020 until 31$^{st}$ March 2021), obtained from the official DGS reports and shared publicly in a specific repository (https://github.com/dssg-pt/covid19pt-data). These data were used to calibrate the values of the first day of each simulation (corresponding to the 23$^{rd}$ March 2020), retrieving the number of official COVID-19 cases per municipality. Afterwards, these data were also used to compare the results and the timelines of the several simulations.

### 2.3. Methodological procedure

**2.3.1. SEIR modeling.** We used a simulation procedure based on a parsimonious SEIR model, adapted from a study developed for Tokyo in Japan, and using R software tools [31]; (https://www.databentobox.com/2020/03/28/covid19_city_sim_seir/). The SEIR model depicts the temporal evolution of disease spread in a sequence of phases, or epidemiological compartments, that represent different conditions. The parameters and conditions applied to the simulations were as follows (Fig 4):

**S**–the **susceptible** population was obtained from the number of residents in each municipality, reported in 2019 [23]. It was assumed that immunity to the virus was inexistent; therefore, the entire population was considered susceptible at the start of the simulations. We considered that once a person was infected, he/she would no longer be susceptible to the virus, mostly by gaining a certain level of immunity and some by eventual death.

**E**–the **exposed** population depends on the probability of contact between infected people and the susceptible population. In the SEIR model, the transition rate between the susceptible and exposed compartments in the model is given by parameter β, or the inverse of the contact period 1/β, representing how often a susceptible-infected contact results in a new exposure.

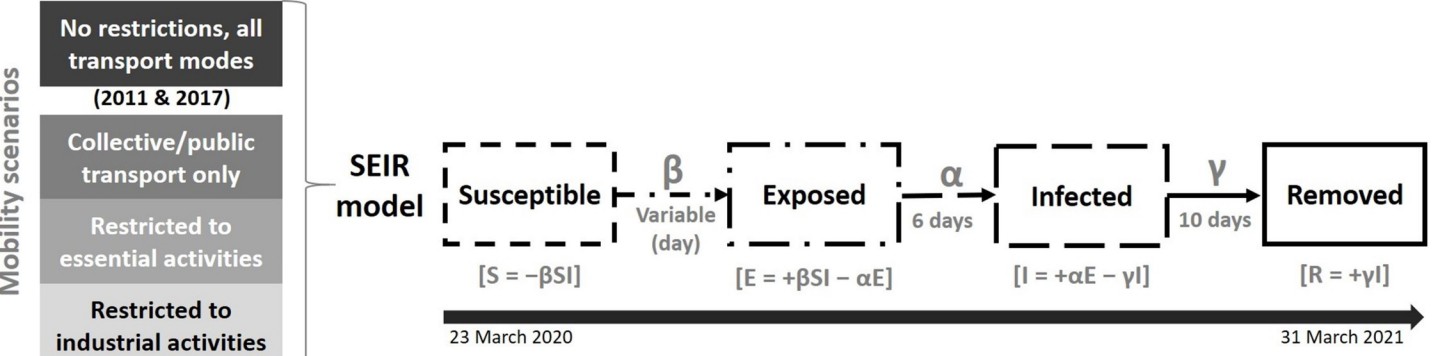

**Fig 4. Schematic representation of the simulation procedure applied.**

We estimated β values from the daily $R_t$ values and the transition rate between infected and removed (parameter γ, see phase R below), considering that $R_t$ is a function of β/γ [32].

**I**–the **infected** population derives from the rate at which the people exposed to the virus become infectious, calculated according to the period of incubation of the virus. Following recent epidemiological information (https://www.ecdc.europa.eu/en/covid-19/latest-evidence/infection; [33], we applied a constant incubation period of 6 days (the inverse of this period is parameter α = 1/6).

**R**–most people **recover** from the virus and stop being infectious after 10 days, according to recent epidemiological results [34] and expressed in the patient isolation rules applied in the country. The inverse of this period gives the last parameter of the SEIR model, or the rate an infected person recovers and moves into the final phase (γ = 1/recovery time of 10 days). This parameter, together with $R_t$, was used to estimate daily β.

**2.3.2. Mobility scenarios.** The parameters α and γ are disease-specific and do not vary spatially; therefore, the same rates were applied to all municipalities in both metropolitan areas. Conversely, the β parameter varies in time and space, and daily values adjusted for each metropolitan area were applied. The variation of β in space, derived from the mobility patterns, is defined by people's flows within each metropolitan area. These flows were obtained from the different commuting levels identified (Table 1 and Fig 4): 1) unrestricted mobility, all transport modes; 2) mobility based on the use of public collective transport; 3) mobility restricted to essential activities; 4) mobility restricted to industrial activities. We created an origin-destination (OD) matrix for each flow type, calculating the number of people traveling within each municipality and between each possible pair or municipalities every day, in LMA and PMA separately. We focused on the potential influence of commuting flows in virus spread and evolution, assuming that the effects of preventive measures, such as physical distancing, hand sanitization, and face masks, were expressed within the β/$R_t$ values applied. The simulated exposed and infected cases for each municipality were calculated by assuming contact between susceptible people, local infected cases and imported infectious cases, which vary with the degree of commuting flows in the mobility scenarios defined (Fig 4). The parameters of the SEIR model were kept constant, except β, and no other variables were integrated, to allow focusing on the analysis of potential differences resulting specifically from different mobility levels.

Based on the flow of people between municipalities and the parameters defined for the SEIR model, we simulated the daily new cases and the active cases of infection per day. We evaluated how they were spatially distributed within each metropolitan area. The simulation started on the 23rd March of 2020 when data disaggregated at the municipal level was first available. The SEIR model was calibrated with the cases recorded for that day, regarding the number of susceptible, exposed, and infected people in each municipality. Afterwards, the model was run for 373 days until 31st March 2021—the period for which $R_t$ values daily were available at the time of the development of the simulations.

Subsequently, we compared the evolution of the number of newly infected by day between all the simulated mobility scenarios in the study period. We analyzed the potential effect of mobility restrictions on the proportion of infected people. Additional simulations for each metropolitan area were obtained for the scenarios representing unrestricted mobility 2011 and mobility restricted to essential activities, using the upper and lower confidence intervals (95%) of the corresponding daily $R_t$ values. The comparison of the simulated values with official records of infected cases was made only for the period between 23rd March and 26th October because, afterwards, the reporting of new cases in Portugal changed to an incidence rate for 14 days, and this format is not suitable to compare with the simulated cases.

**2.3.3. Time-series clustering analysis.** The trends of COVID-19 spread were further explored at the municipal level through time-series clustering analysis, considering the scenarios of unrestricted mobility 2011 and mobility restricted to essential activities. In each metropolitan area, the simulated timelines of the virus spread for each municipality were partitioned into groups, based on the (dis)similarities between the temporal sequences of predicted new daily infections. We used Dynamic Time Warping (DTW) distance as a dissimilarity measure [35], using the package *dtwclust in* R software [36]. This algorithm aims to find the optimum warping path between multiple time series that may vary in speed, allowing their synchronization by applying transformations such as stretching or warping. The following step was to compute time-series prototypes that efficiently represent the most significant features of all series in each cluster, assuming that all time-series within a cluster are self-similar [37]. We used an iterative, global method based on DTW known as DTW barycenter averaging (DBA), which involves the use of one series as a reference (centroid) randomly selected from the dataset, as developed by Petitjean et al. [38]. Finally, we used a hierarchical clustering method because our dataset is relatively small, and it was not required to specify the desired number of clusters previously.

# 3. Results

## 3.1. Estimated new infections (SEIR model) by mobility scenario

In both metropolitan areas and for all mobility scenarios, the simulated values of daily new infections are low until 4th May 2020, subsequently increasing at different rates (Fig 5). It coincides with the first lockdown period, established between 23rd March and 4th May 2020. In LMA, excluding the scenario of unrestricted mobility 2017, the highest daily values of new infections are reached in mid-January 2021. In contrast, in PMA the highest daily values are reached by mid-November 2020 decreasing sharply afterwards, until early January 2021 when values increase again, although at a much lower level than before.

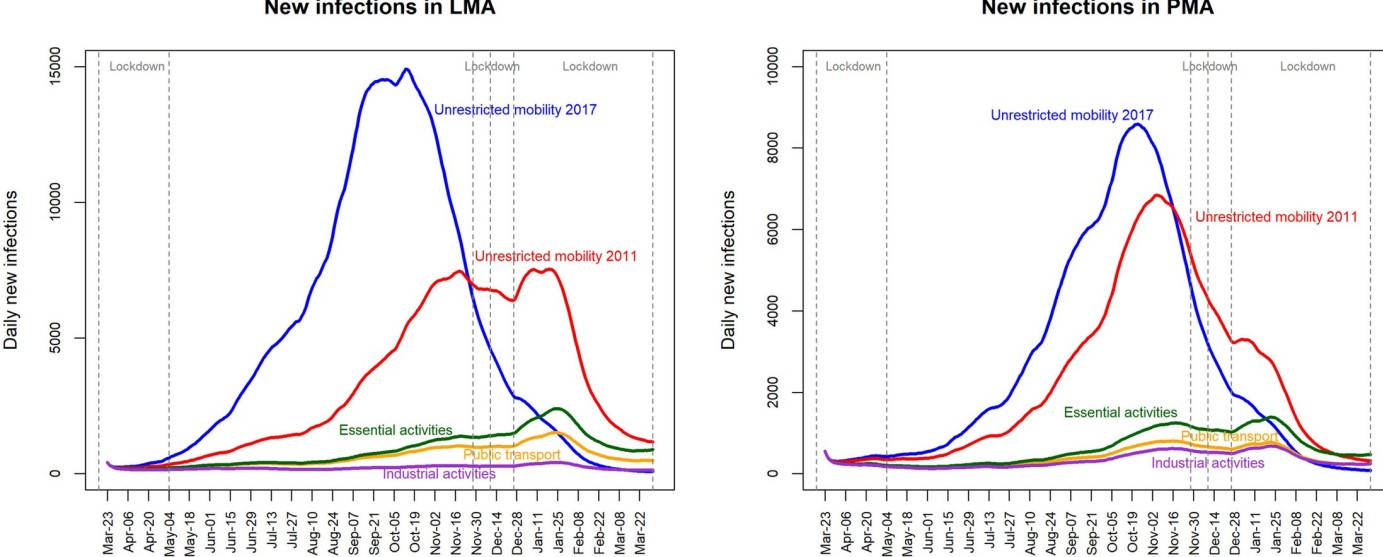

**Fig 5.** Simulated number of daily new infections by mobility scenario (described in Table 1), between 23rd March 2020 and 31st March 2021, in the metropolitan areas of Lisbon (left) and Porto (right). Stricter lockdown periods, with restrictions on intermunicipal mobility, are delimited by the vertical dashed lines and cover the periods between 23rd March and 4th May 2020, 28th November to 8th December 2020, and from 27th December 2020 until 31st March 2021.

In most simulations, the number of infected people in LMA is higher than in PMA. However, these correspond to a lower proportion of the total susceptible population in the metropolitan area (Table 2). An exception is found for the scenario of mobility restricted to industrial activities, whose proportion of the infected population and the total number are greater in PMA. In both areas, the scenario representing unrestricted mobility with 2017 data reached the highest number of infections (Fig 5). For LMA, it is estimated that 64% of the total population would have been infected by the end of March 2021, with a daily maximum value of new infections close to 15 000 people reached by mid-October 2020. For PMA, the proportion of people infected would be 56%, and the daily maximum would be around 8 500 new infections, reached in late October 2020 (Table 2).

In both metropolitan areas, the mobility scenario with the lowest estimated values of infections is the one restricted to industrial activities (Fig 5). In LMA, the timeline of daily new infections is very similar between the scenarios of mobility restricted to essential activities and the public transport use. In contrast, in PMA the strongest similarity is found between mobility scenarios regarding public transport use and industrial activities (Fig 5).

The timeline of official records of new infections follows different mobility scenarios over time and vary between metropolitan areas. In LMA, the daily official records (until 6th July 2020) follow a similar shape to the scenario's timeline restricted to essential activities (Fig 6). In July 2020, coinciding with the change from daily to cumulative weekly records, LMA official records reach the simulated values for the unrestricted mobility 2011 scenario and increase since mid-September 2020 (Fig 6). In PMA, some daily official records of March and April 2020 surpass the upper threshold of the simulated values for the unrestricted mobility 2011, stabilizing in May 2020 close to the timeline of the mobility scenario restricted to essential activities. This trend continues until the end of August 2020, when weekly cumulative records start to follow the timeline given by the upper threshold of the scenario restricted to essential activities (Fig 6). Since 19th October 2020, official weekly records progress similarly to the unrestricted mobility 2011 scenario (Fig 6).

## 3.2. Clusters of municipal trends

We identified different clusters of municipalities according to the trends of COVID-19 spread over time. In LMA, considering the scenario of unrestricted mobility 2011, most municipalities show a progressive increase in daily new infections since mid-September until late November 2020, more evident from mid-October to mid-November. Afterwards, the number of new infections remains relatively high until the end of January 2021, but dispersed over more

**Table 2. Results on infected cases, obtained from the simulations with different mobility scenarios, for LMA and PMA.**

| Study areas | Lisbon Metropolitan Area (LMA) | | | Porto Metropolitan Area (PMA) | | |
|---|---|---|---|---|---|---|
| Mobility scenarios | Total no. infected | % infected | Max no. daily infected | Total no. infected | % infected | Max no. daily infected |
| Unrestricted mobility, all transport modes 2011 | 1 827 526 | 43.4 | 7534 | 836 647 | 48.8 | 6844 |
| | | | [2021-01-19] | | | [2020-11-05] |
| Unrestricted mobility, all transport modes 2017 | 1 808 035 | 64.4 | 14906 | 968 059 | 56.3 | 8588 |
| | | | [2020-10-14] | | | [2020-10-23] |
| Mobility based on the use of public collective transport | 238 320 | 8.5 | 1518 | 152 590 | 8.9 | 805 |
| | | | [2021-01-23] | | | [2020-11-16] |
| Mobility restricted to essential activities | 326 322 | 11.7 | 2397 | 230 095 | 13.6 | 1391 |
| | | | [2021-01-24] | | | [2021-01-22] |
| Mobility restricted to industrial activities | 84 958 | 3.0 | 420 | 126 345 | 7.5 | 681 |
| | | | [2021-01-23] | | | [2021-01-22] |

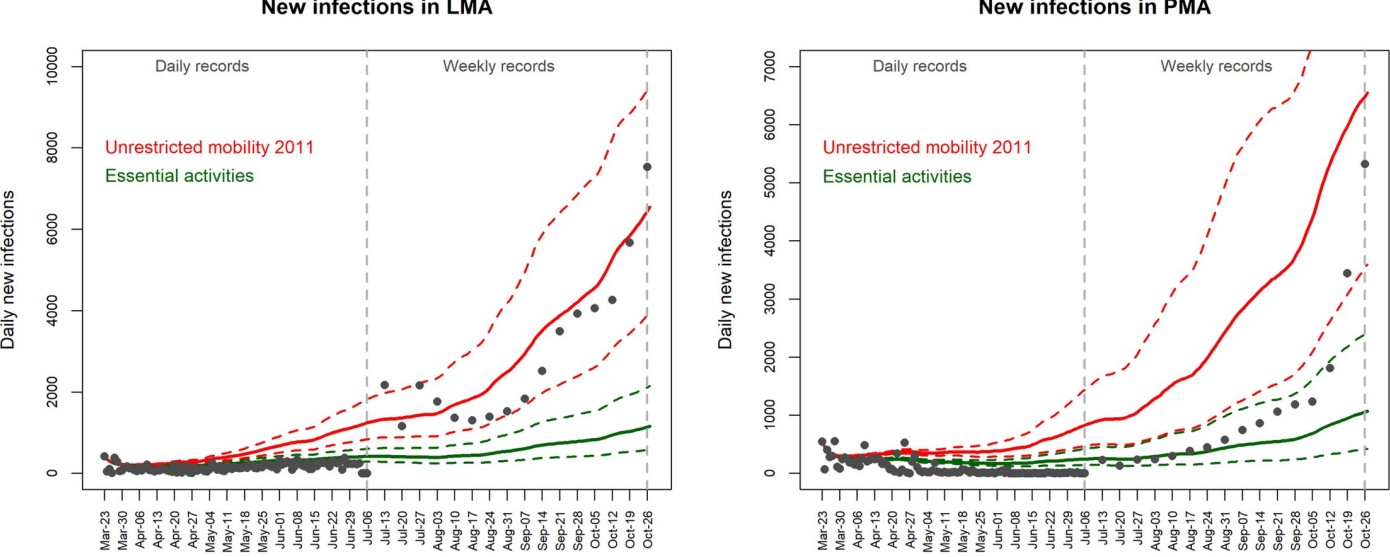

**Fig 6.** Simulated number of daily new infections for the scenarios regarding unrestricted mobility 2011 and mobility restricted to essential activities, between 23rd March and 26th October 2020, in the metropolitan areas of Lisbon (LMA, left) and Porto (PMA, right). Estimated values considering the upper and lower thresholds of the confidence interval (95%) of $R_t$ are represented as dashed lines. Official recorded cases of new infections are represented as points, daily values until the 6th July, and cumulative weekly values afterwards.

municipalities (Fig 7A). Lisbon and Sintra form a specific cluster, with the highest estimated values of daily new infections, although substantially higher for Lisbon. Cascais, Loures, and Oeiras show intermediate values of new infections, forming another cluster; Almada and Amadora come after with regard to estimated new infections and constitute a specific time-series cluster, despite being geographically apart and separated by the river Tagus.

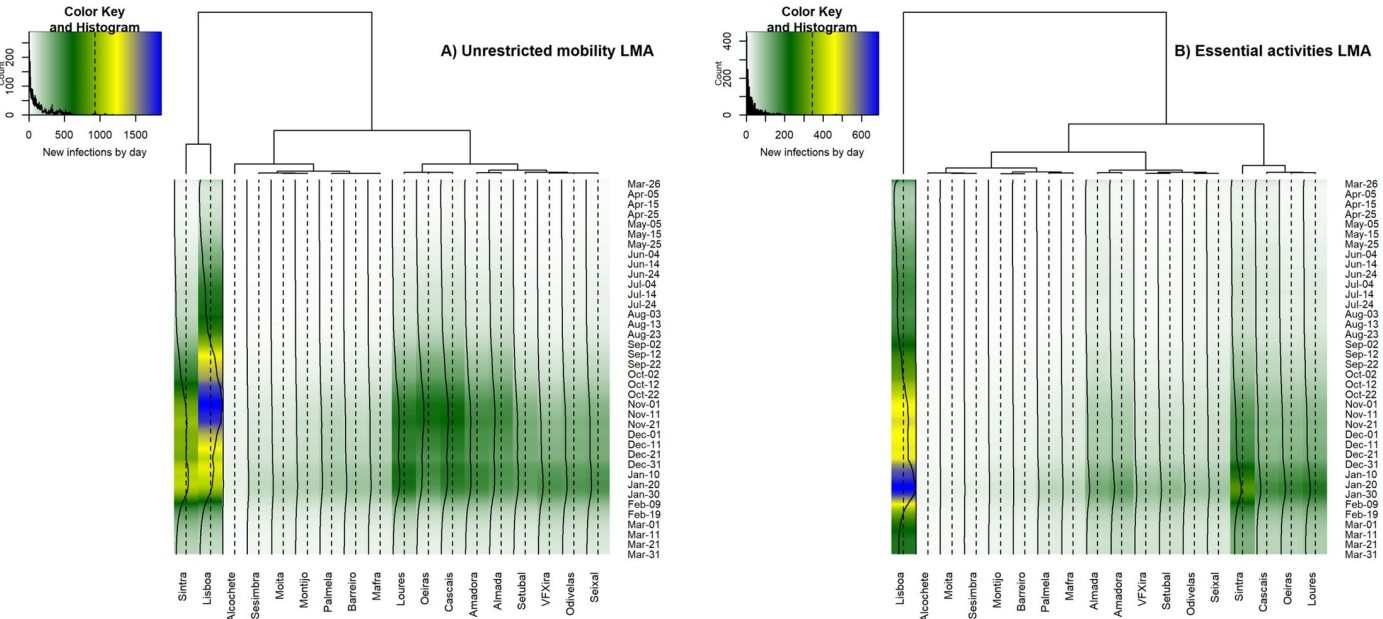

**Fig 7.** Time-series clustering of the daily new infected in the municipalities of LMA, for the simulation regarding the unrestricted mobility of 2011 (A, left) and restricted to essential activities (B, right). The color key is the same for both metropolitan areas and mobility scenarios, to facilitate visual comparisons, but the range of values (new infections by day) differs. Obtained with dtwclust package in R software [37].

In the scenario representing essential activities (Fig 7B), the municipality of Lisbon becomes isolated in a specific cluster, reaching the highest daily values of infections in January 2021. Cascais, Oeiras, and Loures form again a specific cluster with intermediate values of new daily infections; this cluster is then linked to Sintra, the municipality with the second-highest values of daily infections, but much lower than the ones obtained for Lisbon. Almada and Amadora are again considered a specific cluster due to the similarity of simulated values over time. Moita and Sesimbra compose the cluster with the lowest numbers of new infections by day.

In PMA, the overall trends considering unrestricted mobility 2011 show higher numbers of daily new infections between October and November 2020 (Fig 8A). The highest numbers of new infections by day are found in Vila Nova de Gaia (VNGaia), a trend that remains until the end of January 2021. The second highest values are found in Porto, which forms a specific cluster with VNGaia. Matosinhos is the third municipality regarding new daily infections and remains separated from other municipalities until the third level of connection. Santa Maria da Feira and Gondomar present intermediate values of new daily infections and form another cluster. Vila do Conde and Paredes are also strongly connected regarding the evolution of the new daily infections, as well as Santo Tirso and Oliveira de Azemeis (Oliv Azemeis), even though these municipalities are geographically apart from each other. The cluster formed by Arouca and Vale Cambra shows the lowest values.

In the scenario representing essential activities (Fig 8B), the municipality with the highest numbers of daily new infections is Porto, followed by Vila Nova de Gaia, forming a specific cluster with the highest values of new daily infections until the end of January 2021. Santa Maria da Feira (SMFeira) and Matosinhos define the second cluster with higher values, which is then linked to Maia, differing from the spatial groups formed in the unrestricted mobility scenario. Valongo and Gondomar compose the second cluster with higher values, differing from the simulations obtained with the unrestricted mobility scenario. Oliveira de Azemeis

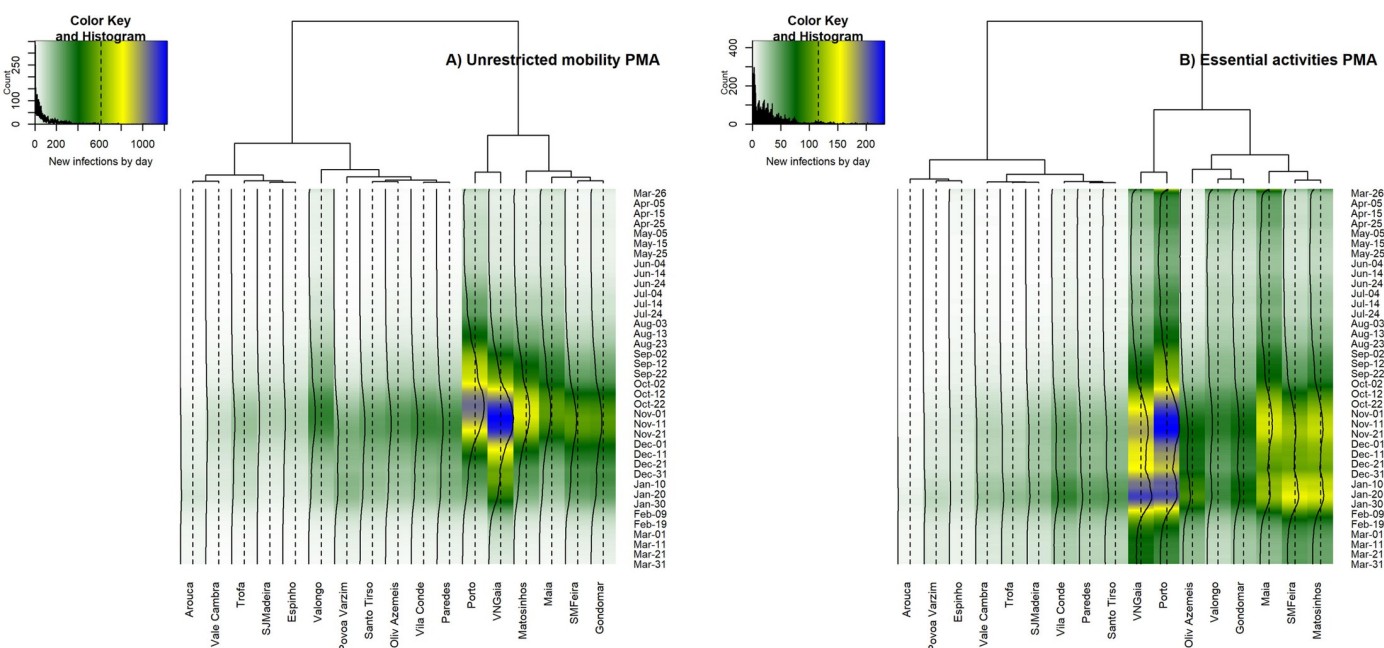

**Fig 8.** Time-series clustering of the daily new infected in the municipalities of PMA, for the simulation regarding the unrestricted mobility of 2011 (A, left) and restricted to essential activities (B, right). The color key is the same for both metropolitan areas and mobility scenarios, to facilitate visual comparisons, but the range of values (new infections by day) differs. Obtained with dtwclust package in R software [37].

shows intermediate values of daily new infections, being linked in a second level to the cluster formed by Valongo and Gondomar.

## 4. Discussion

In a pandemic context, the restrictions imposed on travel and social contact emerge from the need to reduce the contact between humans to restrain the dissemination of the virus, as past epidemic experiences reveal [13, 39]. Based on an adapted SEIR model and official $R_t$ cases recorded at regional scale, we simulated the number of daily new infections in metropolitan areas for different mobility scenarios, to test the potential effect of travel restrictions on virus dissemination. The timelines obtained from the different simulated scenarios indicate that virus spread has responded to the mobility restrictions implemented. The number of new infections decreased after lockdown periods, with a certain delay, a pattern also found by Dehning et al. (2020) [18]. Indeed, along the pandemic period, ongoing for over a year, mobility restrictions have been implemented throughout the world to curb the spread of the disease, with variable effects. In China, Jia et al. (2020) [40] found that mobility accurately predicted the spatiotemporal pattern of infections, by analyzing mobile phone data. In another recent study in Taiwan, Chang et al. (2021) [41] found that intra-city travel restrictions had more impact than intercity ones on outbreak patterns, both in space and time. Our findings show that the mobility flows after the first lockdown in Portugal, implemented from mid-March to early May 2020, never reached the pre-pandemic levels during the year 2020, even when most restrictions were lifted. Other studies corroborate this for the country via the analysis of Google Mobility Data [22], indicating, for most activities, a distance from the mobility baseline even after eight months. This reduction of mobility is not homogeneous amongst activities, as it was found for several countries; in Portugal, retail-recreation and transit stations showed the sharpest decreases in mobility when the first lockdown was implemented, in March 2020, whereas permanency in residence has increased [22]. In India, the sharpest reduction was found for retail and recreation (73.4%), followed by transit stations (66%), workplaces (56.7%), grocery and pharmacy (51.2%), and visits to parks (46.3%) [42].

Our results indicate that until mid-October in PMA, the evolution of virus spread followed closely the timeline obtained with the scenario of mobility restricted to essential activities (between the two CI thresholds), with values of new infections increasing afterwards. In LMA, this change happens earlier, since July, with weekly cumulative records following the timeline given by the unrestricted mobility scenario, at the lower threshold between mid-August and mid-September. These results are consistent with prior analysis done for the country, which evidenced secondary outbreaks in July 2020 especially in the urban and suburban areas of Lisbon, likely due to the ease of restrictions [21]. Afterwards, the increase in new infections may be related to reopening strategies, with a desired "return to normal" encouraged by the government, even though preventive measures (such as the use of a face mask and hand hygiene) were kept. Other studies have also been able to associate the SARS-CoV-2 virus spread with control measures implemented in different countries [18, 43, 44].

The number of infections has drastically changed in January 2021 in Portugal, compared to the previous year. After Christmas, the country went through a challenging period in the number of infected cases and fatalities, and the capacity of the health system to respond was challenged. The simulations captured these conditions, which show a sharp increase in daily new infections shortly after Christmas, particularly in LMA. The number of new infections started to decrease in mid-January 2021, following the implementation of mobility and social restrictions on the 27th December, and a stricter lockdown established on 16th January 2021, including one week later the suspension of all school activities and heavy restrictions on inter-

municipal mobility (DR 3/2021). The lack of strict measures during the Christmas period and the usual family gatherings during this season are likely responsible for this strong increase in virus incidence. The appearance of new variants of the virus, supposedly more contagious, is also to be accounted for [45]. Other reasons may be linked to environmental conditions since the virus dissemination increases when the temperature is lower, although climatic variables alone cannot explain this spread dynamics [46, 47]. Under such circumstances, reopening strategies should take into account the control of mobility flows, the maintenance of remote working activities and social distancing measures, and the cautious restart of social events, as suggested by prior research [48–50].

We also found that the weight of the industrial activities differs between the two metropolitan areas, with LMA dynamics being more supported by services. In contrast, PMA has a higher proportion of industrial activities. The distribution of the metropolitan areas' essential activities is also different—as verified by the time-series clusters, which also show that the timeline and concentration of new infections in the municipalities differ between scenarios. In LMA, Lisbon absorbs most of the flows regardless of the scenario considered, whereas in PMA, the main city (Porto) does not predominate so evidently, although it is part of the cluster with higher number of infections. First studies on the geographical dispersion of COVID-19 in Portugal found that the diffusion of the virus started in densely populated areas with a high concentration of economic activities [51], a pattern likely associated with the intensity of commuting flows. Some differences are also found regarding public transport use; in PMA, public transport mobility seems to be associated with the flow of people working in industrial activities, as the timeline of virus spread is very similar. Instead, in LMA, the public transport flow seems to be linked with people working in essential activities, although this cannot be confirmed due to the lack of connection between the two databases. The link between the COVID-19 pandemic and public transport use has been investigated for different areas. Comparing countries with mobility restrictions in March 2020, such as Portugal, with others that did not follow that path, like the United Kingdom, reveals an overall decay of public transport use by 95% and 75%, respectively [52]. In Poland, a 77% decrease in public transport use was verified [53], and in Wuhan, where the virus first spread, all public transportation services were suspended [54], an extreme measure that is unlikely to be implemented in other countries [52].

The SEIR model used assumed most parameters as constant in the population, a choice that limited the analysis of potential epidemiological differences within the metropolitan areas. Also, no other variables regarding the response of the health system or predictors of human behavior were added. Despite these limitations, the retrospective analysis carried out allowed to focus on the specific effects of mobility levels in virus dissemination, which could help understanding if more tailored strategies can be implemented for each metropolitan area or municipality, according to their particular mobility conditions, and increase the public acceptance of the restrictions implemented.

## 5. Conclusion

Human mobility and transport patterns are associated with disease spread in a population. In this study, we simulated the spread of COVID-19 in metropolitan areas in Portugal for one year, considering several mobility scenarios with different restrictions. The dissemination of the virus varies between metropolitan areas and between their municipalities, indicating different patterns in mobility and transport use. These differences should be considered in reopening strategies in a pandemic context, and integrated in strategies to improve the conditions of metropolitan areas. Mobility patterns will likely change in the metropolitan areas beyond the

duration of the outbreak, with implications to human choices and activities and the overall socio-economic dynamics of these areas. In this context, further work is needed: to understand mobility restrictions' effects on virus spread at different scales; how these restrictions can be integrated in reopening strategies, uncompromising public health; how other social and hygiene measures can be of complement; and how the mobility patterns of metropolitan areas can be improved.

## Author Contributions

**Conceptualization:** Jorge Rocha.

**Data curation:** Sandra Oliveira.

**Formal analysis:** Sandra Oliveira.

**Methodology:** Ana Isabel Ribeiro, Paulo Nogueira, Jorge Rocha.

**Software:** Sandra Oliveira.

**Supervision:** Jorge Rocha.

**Visualization:** Ana Isabel Ribeiro.

**Writing – original draft:** Sandra Oliveira.

**Writing – review & editing:** Ana Isabel Ribeiro, Paulo Nogueira, Jorge Rocha.

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
