## [Decision Letter · Decision Letter 0]

2 Mar 2022

PONE-D-21-30815

Simulating the effects of mobility restrictions in the spread of SARS-CoV-2 in metropolitan areas in Portugal

PLOS ONE

Dear Dr. Oliveira,

Thank you for submitting your manuscript to PLOS ONE. After careful consideration, we feel that it has merit but does not fully meet PLOS ONE’s publication criteria as it currently stands. Therefore, we invite you to submit a revised version of the manuscript that addresses the points raised during the review process.

We look forward to receiving your revised manuscript.

Kind regards,

Vanessa Carels, Staff Editor

on behalf of 

Muhammad Qasim, Ph.D

Academic Editor

PLOS ONE

Journal Requirements:

Additional Editor Comments (if provided):

Reviewers' comments:

Reviewer's Responses to Questions

**Comments to the Author**

1. Is the manuscript technically sound, and do the data support the conclusions?

Reviewer #1: Yes

Reviewer #2: Yes

2. Has the statistical analysis been performed appropriately and rigorously? 

Reviewer #1: Yes

Reviewer #2: Yes

3. Have the authors made all data underlying the findings in their manuscript fully available?

Reviewer #1: Yes

Reviewer #2: Yes

4. Is the manuscript presented in an intelligible fashion and written in standard English?

Reviewer #1: Yes

Reviewer #2: Yes

5. Review Comments to the Author

Reviewer #1: the manuscript titled "Simulating the effects of mobility restrictions in the spread of SARS-1 CoV-2 in metropolitan

2 areas in Portugal", is well-presented. But there are few suggestions to be considered.

1. there are few grammatical and Syntax errors.

2. the quality of all figures especially 3 to 6 should be improved.

3. Conclusion should be more concise.

4. Similarly, abstract should be more focused on the study and its outcome/results. less background information will be enough.

Reviewer #2: In this manuscript the authors simulate the COVID restrictions and their effect on the spread of SARS-CoV-2 in two metropolitan cities Lisbon and Porto of Portugal. The study was based on SEIR model to estimate new daily infections. Four different mobility scenarios ranging from full restrictions to unrestricted full activity, were selected/simulated. The study has practical implications in determining/estimating the future number of infections as well as accepting the restriction practices in masses. The study has a strong hypothesis, the methodology is strong and results are presented in an intelligent fashion.

6. PLOS authors have the option to publish the peer review history of their article (what does this mean?). If published, this will include your full peer review and any attached files.

Reviewer #1: **Yes: **Saba khaliq

Reviewer #2: No

---

## [Author Response · Author response to Decision Letter 0]

20 Apr 2022

Response to reviewers:

We thank the reviewers for the comments and suggestions provided. We respond to each point raised below.

Reviewer #1: 

The manuscript titled "Simulating the effects of mobility restrictions in the spread of SARS-1 CoV-2 in metropolitan 2 areas in Portugal", is well-presented. But there are few suggestions to be considered.

1. there are few grammatical and Syntax errors.

R1: We have revised the manuscript and tried to correct any grammatical and syntax errors. Some changes were made throughout the text, please see revised version of the manuscript.

2. the quality of all figures especially 3 to 6 should be improved.

R2: We understand the point raised and we have revised the figures as requested, improving the image quality. Please see revised figures 3 to 6.

3. Conclusion should be more concise.

R3: Conclusion was revised accordingly, we have removed nonessential text. Please see revised version of the manuscript (page 13).

4. Similarly, abstract should be more focused on the study and its outcome/results. less background information will be enough.

R4: As suggested, the abstract was revised to be more focused and we have removed background information that was not essential in this section. Please see the revised version of the manuscript, abstract section (page 1). 

Reviewer #2: 

In this manuscript the authors simulate the COVID restrictions and their effect on the spread of SARS-CoV-2 in two metropolitan cities Lisbon and Porto of Portugal. The study was based on SEIR model to estimate new daily infections. Four different mobility scenarios ranging from full restrictions to unrestricted full activity, were selected/simulated. The study has practical implications in determining/estimating the future number of infections as well as accepting the restriction practices in masses. The study has a strong hypothesis, the methodology is strong and results are presented in an intelligent fashion.

R5: We thank the reviewer for the positive feedback. The revised version of the manuscript incorporates some changes intended to further improve its quality.

---

## [Decision Letter · Decision Letter 1]

7 Jul 2022

PONE-D-21-30815R1Simulating the effects of mobility restrictions in the spread of SARS-CoV-2 in metropolitan areas in PortugalPLOS ONE

Dear Dr. Oliveira,

Thank you for submitting your manuscript to PLOS ONE. After careful consideration, we feel that it has merit but does not fully meet PLOS ONE’s publication criteria as it currently stands. Therefore, we invite you to submit a revised version of the manuscript that addresses the points raised during the review process.

We look forward to receiving your revised manuscript.

Kind regards,

Muhammad Qasim, Ph.D

Academic Editor

PLOS ONE

Journal Requirements:

Additional Editor Comments:

the manuscript titled "Simulating the effects of mobility restrictions in the spread of SARS-1 CoV-2 in metropolitan

2 areas in Portugal", is well-presented. But there are few suggestions to be considered.

1. there are few grammatical and Syntax errors.

2. the quality of all figures especially 3 to 6 should be improved.

3. Conclusion should be more concise.

4. Similarly, abstract should be more focused on the study and its outcome/results. less background information will be enough.

Reviewers' comments:

Reviewer's Responses to Questions

**Comments to the Author**

1. If the authors have adequately addressed your comments raised in a previous round of review and you feel that this manuscript is now acceptable for publication, you may indicate that here to bypass the “Comments to the Author” section, enter your conflict of interest statement in the “Confidential to Editor” section, and submit your "Accept" recommendation.

Reviewer #3: (No Response)

Reviewer #4: All comments have been addressed

2. Is the manuscript technically sound, and do the data support the conclusions?

Reviewer #3: Yes

Reviewer #4: Yes

3. Has the statistical analysis been performed appropriately and rigorously? 

Reviewer #3: No

Reviewer #4: Yes

4. Have the authors made all data underlying the findings in their manuscript fully available?

Reviewer #3: Yes

Reviewer #4: Yes

5. Is the manuscript presented in an intelligible fashion and written in standard English?

Reviewer #3: Yes

Reviewer #4: Yes

6. Review Comments to the Author

Reviewer #3: The manuscript "Simulating the effects of mobility restrictions in the spread of SARS-CoV-2 in metropolitan areas in Portugal" is well presented. The results are described in good fashion and can be implicated in COVID-19 restriction policies.

Does it need more statistical evaluations along with (SEIR model)?

Reviewer #4: Authors ahs addressed all the suggested changes, the manuscript can be accepted after fulfillment of Journals requirements.

7. PLOS authors have the option to publish the peer review history of their article (what does this mean?). If published, this will include your full peer review and any attached files.

Reviewer #3: No

Reviewer #4: **Yes: **Saba khaliq

---

## [Author Response · Author response to Decision Letter 1]

27 Jul 2022

Review Comments to the Author

Reviewer #3: The manuscript "Simulating the effects of mobility restrictions in the spread of SARS-CoV-2 in metropolitan areas in Portugal" is well presented. The results are described in good fashion and can be implicated in COVID-19 restriction policies.

Does it need more statistical evaluations along with (SEIR model)?

R5: We thank the reviewer for the overall positive feedback. 

Regarding the suggestion for more statistical evaluations along with SEIR model, we would like to enhance that the statistical analysis done had to be adapted to the data available. As mentioned in the manuscript, the number of daily infections are not available at municipal level (Lines 280 and following in the revised version with track changes), therefore no further comparison with the simulations could be done. Another limitation was that the data of people working in essential activities is not connected to the commuting data nor transport type, therefore they had to be used as separate scenarios (Table 1). Considering the characteristics (limitations) of the data and the objectives of the study, we have then decided to apply a time-series clustering analysis method (section 3.2.) which, to our knowledge, had not yet been used in this context, to obtain further understanding of the patterns and trends of the resulting simulations, that could not be obtained otherwise.

Reviewer #4: Authors have addressed all the suggested changes, the manuscript can be accepted after fulfillment of Journals requirements.

R6: We thank the reviewer for the positive feedback.

---

## [Decision Letter · Decision Letter 2]

25 Aug 2022

Simulating the effects of mobility restrictions in the spread of SARS-CoV-2 in metropolitan areas in Portugal

PONE-D-21-30815R2

Dear Dr. Oliveira,

We’re pleased to inform you that your manuscript has been judged scientifically suitable for publication and will be formally accepted for publication once it meets all outstanding technical requirements.

Kind regards,

Hana Maria Dobrovolny, Ph.D

Academic Editor

PLOS ONE

Additional Editor Comments (optional):

Reviewers' comments:

Reviewer's Responses to Questions

**Comments to the Author**

1. If the authors have adequately addressed your comments raised in a previous round of review and you feel that this manuscript is now acceptable for publication, you may indicate that here to bypass the “Comments to the Author” section, enter your conflict of interest statement in the “Confidential to Editor” section, and submit your "Accept" recommendation.

Reviewer #3: All comments have been addressed

Reviewer #4: All comments have been addressed

2. Is the manuscript technically sound, and do the data support the conclusions?

Reviewer #3: Yes

Reviewer #4: Yes

3. Has the statistical analysis been performed appropriately and rigorously? 

Reviewer #3: Yes

Reviewer #4: Yes

4. Have the authors made all data underlying the findings in their manuscript fully available?

Reviewer #3: Yes

Reviewer #4: Yes

5. Is the manuscript presented in an intelligible fashion and written in standard English?

Reviewer #3: Yes

Reviewer #4: Yes

6. Review Comments to the Author

Reviewer #3: The manuscript is good now to be published. The suggestion I raised have been appropriately answered and adjusted.

Reviewer #4: the authors has incorporated all the suggested changes with additional statistical analysis. Moreover, limitations of the current study is also added in this manuscript.

7. PLOS authors have the option to publish the peer review history of their article (what does this mean?). If published, this will include your full peer review and any attached files.

Reviewer #3: No

Reviewer #4: **Yes: **Saba Khaliq

---

## [Editor Report · Acceptance letter]

30 Aug 2022

PONE-D-21-30815R2 

Simulating the effects of mobility restrictions in the spread of SARS-CoV-2 in metropolitan areas in Portugal 

Dear Dr. Oliveira:

I'm pleased to inform you that your manuscript has been deemed suitable for publication in PLOS ONE. Congratulations! Your manuscript is now with our production department. 

Kind regards, 

on behalf of

Dr. Hana Maria Dobrovolny 

Academic Editor

PLOS ONE